# Role of NS1 and TLR3 in Pathogenesis and Immunity of WNV

**DOI:** 10.3390/v11070603

**Published:** 2019-07-03

**Authors:** Sameera Patel, Alessandro Sinigaglia, Luisa Barzon, Matteo Fassan, Florian Sparber, Salome LeibundGut-Landmann, Mathias Ackermann

**Affiliations:** 1Institute of Virology, Vetsuisse Faculty, University of Zurich, CH-8057 Zurich, Switzerland; 2Department of Molecular Medicine, University of Padova, I-35121 Padova, Italy; 3Department of Medicine, University of Padova, I-35128 Padova, Italy; 4Section of Immunology, Vetsuisse Faculty, University of Zurich, CH-8057 Zurich, Switzerland

**Keywords:** NS1, TLR3, West Nile Virus, immunity, pathogenesis, subunit vaccine, West Nile neuroinvasive disease

## Abstract

West Nile Virus (WNV) is a mosquito-transmitted flavivirus which causes encephalitis especially in elderly and immunocompromised individuals. Previous studies have suggested the protective role of the Toll-like receptor 3 (TLR3) pathway against WNV entry into the brain, while the WNV non-structural protein 1 (NS1) interferes with the TLR3 signaling pathway, besides being a component of viral genome replication machinery. In this study, we investigated whether immunization with NS1 could protect against WNV neuroinvasion in the context of TLR3 deficiency. We immunized mice with either an intact or deleted TLR3 system (TLR3KO) with WNV envelope glycoprotein (gE) protein, NS1, or a combination of gE and NS1. Immunization with gE or gE/NS1, but not with NS1 alone, induced WNV neutralizing antibodies and protected against WNV brain invasion and inflammation. The presence of intact TLR3 signaling had no apparent effect on WNV brain invasion. However, mock-immunized TLR3KO mice had higher inflammatory cell invasion upon WNV brain infection than NS1-immunized TLR3KO mice and wild type mice. Thus, immunization against NS1 may reduce brain inflammation in a context of TLR3 signaling deficiency.

## 1. Introduction

West Nile Virus (WNV) is a neurotropic flavivirus that naturally cycles between wild birds and *Culex* spp. mosquitoes which may incidentally transmit the virus to humans and other mammals [1]. Importantly, the virus can cause a fatal neurological disease in humans and horses, although only a minor percentage of infected individuals will show any disease symptoms. In addition, the virus may cause disease and mortality in several bird species [2,3,4]. Vaccines to protect horses and domestic birds have been in use for a long time. In contrast, a vaccine for humans has not yet been licensed, although a variety of vaccine candidates have been proposed, including vaccines based on inactivated viruses, recombinant viruses, chimeric viruses, recombinant DNA, purified proteins, or combinations thereof [5,6,7,8,9,10,11,12,13,14]. Such candidate vaccines have been extensively tested, mostly in mouse models but also in Syrian hamsters [13,15,16,17,18,19,20,21,22,23,24,25] and nonhuman primates [26].

WNV envelope glycoprotein (gE) is a key component of anti-WNV vaccines since immunization with gE confers protective humoral immunity against WNV [13,14,18,24,27,28,29]. WNV gE is the major receptor-binding protein and comprises the epitopes for neutralizing antibodies that map to a region termed domain III (aa 299–400; indicated in yellow in Figure 1) [30]. Besides gE, additional viral antigens have been included in anti-WNV vaccines, such as the membrane (prM) and the capsid (C) proteins to generate virus-like particles, and nonstructural proteins, such as the NS1 glycoprotein (NS1) (indicated in gray in Figure 1).

Including NS1 into vaccines could be a promising approach in particular to provide additional protection for individuals with weaning immune responses against gE or with defects in the Toll-like receptor 3 (TLR3) system. NS1 is a multifunctional protein that is both associated with cell membranes and secreted in the bloodstream as a hexamer. Besides being an essential component of the viral replication complex [31], WNV NS1 is involved in neuroinvasion and evasion of the host immune response [32]. WNV NS1 has been demonstrated to antagonize TLR3 signaling and thus the establishment of a TLR3-dependent antiviral state in neurons [33], although this finding was not observed in another study [34]. WNV NS1 has also been reported to antagonize complement activation [23] and to selectively bind brain endothelial cells resulting in endothelial hyper-permeability and disruption of the blood–brain barrier [35]. Immunization of mice against NS1 has been shown to provide protection against lethal infection with neurovirulent WNV strains [36], but NS1 has also shown poor immunogenicity in some animal models [37]. Data on the role of the TLR3 pathway in WNV protection or pathogenesis are also conflicting. While some studies suggested a protective role of the TLR3 system against WNV [38] and its antagonizing by NS1, important for WNV virulence [33], others have demonstrated that the TLR3 system is required for allowing WNV entry into the brain, thus representing a preconditional system for the development of lethal encephalitis [39]. Similarly, others have linked the increased frequency of WNV encephalitis in elderly humans to increased TLR3 signaling, which may lead to elevated cytokine levels to contribute to the permeability of the blood–brain barrier [40].

Many studies on WNV pathogenesis and immunity focus on highly virulent American strains, such as the cluster of WNV NY99 isolates [16,39,41,42]. In this study we used the ITA09 strain, which was isolated from an asymptomatic blood donor in Italy but was still neurovirulent in the mouse model [43,44]. This strain is representative of WNV lineage 1, which circulates in Western Europe and the Mediterranean area [45].

With these considerations and partly conflicting findings in mind, we set out to address the role of immunization against WNV NS1 in mice with either an intact or else genetically deleted TLR3 system. For this purpose, we produced recombinant gE and NS1 proteins and immunized wild type (WT) mice and congenic TLR3-knockout mice (TLR3KO) with either gE alone or NS1 alone or gE in combination with NS1, followed by challenge infection with WNV ITA09. Based on survival, clinical WNV symptoms, measurements of WNV viremia, serostatus against gE and/or NS1, WNV detection in the brain, measurement of inflammatory cytokines, and immune cell invasion into the brain in response to WNV, we found that neither the TLR3 pathway nor antibodies against NS1 had a principal effect on WNV brain invasion. However, the degree of cellular infiltration in the brain was less severe in WT mice than in TLR3KO mice. Yet, in comparison to non-immunized siblings, TLR3KO mice immunized against WNV NS1 showed reduced cell infiltration to the WNV-infected brain, which was comparable to the degree observed in WT mice with or without previous vaccination against NS1.

## 2. Materials and Methods

### 2.1. Ethics Statement

Ethical justification of the study as well as animal care, use, and termination criteria were approved by the Cantonal Veterinary Office (Zurich, Switzerland) under license number ZH57/2014 (Approval date: 5 February 2014). Housing and experimental procedures were in strict accordance with Swiss animal protection law and in conformity with the European Convention for the Protection of Vertebrate Animals used for Experimental Purposes.

### 2.2. Construction of Recombinant Baculoviruses for Expression of WNV Glycoprotein E- and NS1-Fragments

The nucleotide sequences encoding the external fragments (ectodomains without stop codons) of WNV glycoproteins E and NS1 (Figure 1), respectively, were codon-optimized for expression in insect cells. Moreover, two variants of each construct were made, one with a C-terminal c-myc tag without a stop codon, the other with two C-terminal tags (V5 and 6xhis followed by a stop codon) before they were commercially synthesized and cloned into the BamHI restriction enzyme site of the pUC57 vector (GenScript, Piscataway, NJ, USA). The synthetic constructs were excised by BamHI digestion to be cloned into the BamHI site of a pDONR221-derived Gateway vector, which comprised an upstream synthetic baculovirus signal sequence as well as a downstream GST tag prior to the stop codon and which provided attL1 and attL2 sequences for targeted recombination into the pDEST8 vector by means of the Gateway LR reaction (Life Technologies, Carlsbad, CA, USA). The resulting pDEST8 constructs were sequenced to verify the integrity and orientation of the inserts before being transposed into baculovirus bacmids (Life Technologies’ Bac-to-Bac Baculovirus Expression System). The inserts of the bacmids were again verified by sequencing before the bacmids were used to transfect Sf9 cells in order to reconstitute the desired recombinant baculoviruses.

### 2.3. Virus Strains and Cell Lines

All work with infectious WNV was done under biosafety level 3 (BSL-3) conditions. WNV Italy 09 (ITA09) is a WNV lineage 1 strain which was isolated in Vero cells from a viremic blood donor, as reported [44]. The WNV ITA09 stock used for experiments was at passage 2. WNV Egypt-101 (Eg-101, kindly provided by the OIE Reference Centre for West Nile Disease, Istituto Zooprofilattico Sperimentale “G. Caporale”, Teramo, Italy) is a non-lethal WNV lineage 1 strain which was used for neutralization assays. Vero cells (ATCC CCL81™; American Type Culture Collection, Manassas, VA, USA), an African Green Monkey kidney-derived cell line, were used for WNV expansion, titration, and neutralization assays. Vero cells were grown in Dulbecco modified Eagle medium (DMEM), supplemented with 10% fetal bovine serum (FBS). *Spodoptera frugiperda*-derived Sf9 and Mimic Sf9 were obtained from Thermo Fisher Scientific (Reinach, BL, Switzerland) and maintained as monolayers at 27 °C. Conventional Sf9 cells were cultured in Grace’s insect medium supplemented with 10% FCS and used for transfections, viral stock production, and ELISA antigen production. The Mimic SF9 cells were cultured in serum-free SF900 III medium (Thermo Fisher Scientific) and used for expression of immunizing antigens.

### 2.4. Animals and Protection Study

C57BL/6J, BALB/c, and B6;129S1-Tlr3tm1Flv/J (TLR3 knockout; TLR3KO) female mice were used for animal experiments. Female WT C57BL/6 mice (6–8 weeks old), and BALB/c mice (6–8 weeks old) were purchased from Charles River (Germany) and housed at the Institute of Laboratory Animal Sciences. TLR3KO mice (6–8 weeks old) were bred in-house. Table 1 below provides the names designated for the groups used in the protection study.

In the protection study (Table 1), 6–8 weeks old female WT C57BL/6 and TLR3KO mice (*n* = 5) were immunized subcutaneously (s.c.) thrice with either 5 µg of gE, NS1, or gE+NS1 supplemented with 10 µg of Quil-A adjuvant or 1/10th of a horse dose of Equip WNV commercial vaccine (Zoetis, Delémont, Switzerland), used as a positive control for the immunization. Negative control mice were vaccinated with protein elution buffer supplemented with 10 µg Quil-A adjuvant. Sera samples after the final booster immunization were tested for neutralizing antibodies and gE or NS1 specific antibodies. All the groups were then inoculated with 10^5^ plaque forming units (pfu) of WNV lineage 1 ITA09 strain intraperitoneally (i.p.). Mice were scored twice daily for signs of infection and euthanized 20 days after infection or earlier depending on whether they reached the end point score. Blood samples were collected at days 2 and 4 post inoculation to measure the viral burden in the plasma. Brain and spleen samples from all the mice were collected after euthanasia for various analyses.

### 2.5. Clinical Scoring

As described previously [13], the clinical signs of WNV infection in mice were subdivided into three categories: loss of body weight, pain-associated behavior, and disease-related physical changes. Table 2 summarizes our scoring system.

Prior to the WNV challenge, all the animals were checked for two weeks and had a score of 0 in all three categories. After WNV inoculation, the animals were scored twice each day for all the traits.

Termination criteria: an animal was terminated by euthanasia if it reached a combined score of 6 or more from all categories or if it reached a score of 3 either in the category loss-of-body-weight or in the category pain-associated-behavior.

### 2.6. ELISA for gE and NS1 Specific Antibodies

First, glutathione-conjugated casein (kindly provided by Dr. Kurt Tobler, University of Zurich, Switzerland) at 1:500 in coating buffer (10 mM Na_2_CO_3_, 40 mM NaHCO_3_, pH9.6) was incubated over night at 4 °C in flat-bottom ELISA plates (Fisher Scientific AG, Wohlen, Switzerland). Then, excess antigen-binding sites were blocked with blocking solution (2% milk powder in phosphate-buffered saline (PBS), supplemented with Tween 20, pH 7.4 (PBST). In the next step, GST-fusion proteins (separate plates for gE and NS1, respectively) were allowed to bind as ELISA antigens to the glutathione molecules in a native manner. For this purpose, the fusion proteins were diluted in blocking solution in a manner to provide an optical density at 450 nm (OD450) of 1.0, upon analysis with the c-myc monoclonal antibody (Life technologies, Carlsbad, CA, USA). Now, mouse sera, prediluted in blocking buffer, were incubated with the antigens. Irrespective of the vaccinating antigen(s), each serum was tested against both NS1- and gE-antigens in order to control for unspecific reactions. Moreover, each serum was tested in two separate dilutions, namely at 1:1000 and at 1:100. Then, goat anti-mouse-horseradish peroxisase (HRP)-conjugated secondary antibody (GE Healthcare, Opfikon, Switzerland) diluted in blocking buffer at 1:1000 was added. Finally, 3,3′,5,5′-tetramethylbenzidine (TMB) substrate (Thermo Scientific, Basel, Switzerland) was prepared according to the manufacturer’s protocol and 100 μL were added per well. The reactions were left at room temperature and stopped after 30 min using an equal volume of 2 M H_2_SO_4_, 10 min before the absorbance was measured at 450 nm. All volumes were 50 µL and all incubations were at 37 °C for one hour, except where stated otherwise. Three washings with 300 µL PBST were done between each step. The final wash before addition of the substrate solution was done with plain PBS after five times washing with PBST.

### 2.7. WNV Neutralization Test

Neutralizing antibodies in mouse sera were titrated by plaque reduction neutralization assay (PRNT) [46]. In brief, two-fold serial dilutions of each serum (from 1:80 to 1:2560) were mixed with an equal volume of medium containing approximately 50–60 pfu of WNV lineage 1 (Eg101 strain) and incubated for 75 min at 37 °C before inoculation onto sub-confluent monolayers of Vero cells (about 5 × 10^5^ cells/well in 6-well plates). After adsorption for 1 h at 37 °C, the inoculum was removed and the cells were overlaid with a gel made of 50% 2× minimal essential medium (2× MEM) medium (with 4% FBS) and 50% agar noble 2% *w*/*v* in sterilized water and incubated at 37 °C, 5% CO_2_. After 3 days, a secondary overlay prepared as above, but with the addition of Neutral Red stain, was added to the well, and on the fourth day plaques were detected as non-stained spots on the red background. The neutralizing antibody titer was calculated as the reciprocal of the serum dilution at which a 50% reduction of plaques (PRNT50) was observed compared to an inoculated control well incubated in the absence of the test serum.

### 2.8. WNV RNA Detection in Plasma and Tissue Samples

For WNV RNA detection in plasma, brain, and spleen samples, total nucleic acids were extracted from samples by using the MagNA Pure 96 SV nucleic acid purification kit (Roche, Basel, Switzerland). For WNV RNA quantification, the NS5 sequence was targeted by real-time RT-PCR using a One Step Real Time kit (Thermo Fisher Scientific, Waltham, MA, USA) as previously described [47]. Real time reactions were run on a 7900 HT Sequence Detection System instrument (Thermo Fisher Scientific).

### 2.9. Cytokine Quantification

Brain tissue samples were immersed in 1 mL of RLT buffer and stored at –80 °C until further processing. The samples were homogenized using an 18G needle and then centrifuged at 12,000× *g* for 10 min. Total RNA was extracted from 200 µL of the total sample using a qiashredder (Qiagen AG, Basel, Switzerland) and an RNeasy mini kit (Qiagen AG, Basel, Switzerland) according to the manufacturer’s protocol. 500 ng of total RNA was reverse transcribed using a QuantiTect Reverse Transcription kit (Qiagen AG, Basel, Switzerland) as per the protocol. Following reverse transcription, cytokine expression in the brain was determined by conducting quantitative SYBR green-based real time RT-PCR (real time qRT-PCR). The reaction mixture contained 2 µL of template cDNA (dilute 1 in 10), 10 µL of 2× SYBR green mix, and 500 nM of primers at a final volume of 20 µL. The initial activation step for the HotStarTaq DNA polymerase was performed at 95 °C for 15 min followed by 45 cycles of 95 °C for 15 s, 60 °C for 30 s, and 72 °C for 30 s, and an extension cycle at 72 °C for 5 min. The real time RT-PCR was carried out using the QuantStudio 7 Flex system. A control sample containing no template was run with each assay and all the reactions were performed in duplicate to ensure reproducibility. The authenticity of the amplified product was determined by melting curve analysis. Relative expression of each target gene was calculated with the ΔΔCt method using β-actin as a house-keeping gene. The following primers were used: β-actin (fwd: 5′-TGG AAT CCC TGT GGG ACC ATG AAA C-3′; rev: 5′-TAA AAC GCA GCT CAG TAA CAG TCC G-3′), TNFα (fwd: 5′-CGT CGT AGC AAA CCA CCA AG-3′; rev: 5′-TTG AAG AGA ACC TGG GAG TAG ACA-3′) and IFNγ (fwd: 5′-GCT CTG AGA CAA TGA ACG CT-3′; rev: 5′-AAA GAG ATA ATC TGG CTC TGC-3′).

### 2.10. Isolation of Leukocytes from Brain Tissues

Half of the brain was shortly rinsed in PBS and minced finely with a scalpel and a quarter of it was transferred into a tube containing 1.5 mL of digestion buffer (RPMI with 10% FCS, 2 mM HEPES, 0.4 mg/mL Collagenase type IV, and 2 mg/mL DNaseI). The sample was incubated at 37 °C. After 45 min, the reaction was stopped with 0.5 M EDTA. The sample was then homogenized with an 18G × 40 mm needle and filtered using a 70 μm cell strainer and centrifuged in a 1.5 mL Eppendorf tube at 450× *g* for 8 min at 4 °C. The cell pellet was washed with 1 mL of cold PBS and centrifuged at 450× *g* for 8 min at 4 °C. Supernatant was discarded. The cell pellet was resuspended in 1 mL 30% Percoll and centrifuged at 12,000× *g* for 30 min at 4 °C. The upper layer containing myelin was removed and 500 µL of cold PBS was added to the tube. The Percoll suspension was filtered using a 70 μm cell strainer and centrifuged at 450× *g* for 8 min at 4 °C. The cell pellet was washed once with PBS and then twice in 1 mL FACS buffer (PBS supplemented with 1% FCS, 5 mM EDTA, and 0.02% NaN_3_) and centrifuged at 450× *g* for 8 min at 4 °C.

### 2.11. Analysis of Leukocyte Infiltration into the CNS by Flow Cytometry

The following antibodies were used: LIVE/DEAD Fixable Near-IR dead cell stain (Life Technologies), CD45:PB (Biolegend, clone 104), CD11B:PE-Cy7 (Biolegend, clone M1/70), Ly6G:FITC (Biolegend, clone 1A8), Ly6C:BV510 (Biolegend, clone HK1.4), and CD45.2:Alexa700 (Biolegend, clone 104). Cells were resuspended in 100 µL of antibody mix and incubated for 30 min at 4 °C in the dark and then washed with FACS buffer and spun down at 450× *g* for 8 min at 4 °C. Cells were fixed in 5% paraformaldehyde for 15 min and then washed twice with FACS buffer and centrifuged at 450× *g* for 8 min at 4 °C. The cells were finally resuspended in 100 µL of FACS buffer. All the data were collected using a Gallios flow cytometer (Beckman Coulter) and analyzed using FlowJo software version 10 (FlowJo LLC). The gating of the flow cytometric data was performed according to the guidelines for the use of flow cytometry and cell sorting in immunological studies [48], including pre-gating on viable and single cells for analysis. Absolute cell numbers of each cell population were calculated based on a defined number of counting beads (BD Bioscience, Calibrite beads) which were added to the samples before the flow cytometric acquisition.

### 2.12. Histology of Brain Samples

Histological analysis of brain samples was performed after hematoxylin and eosin staining. The severity of brain lesions was scored as follows: negative (0), mild lesions possibly associated with viral infection (1), moderate lesions (2), and severe lesions (3).

### 2.13. Statistical Analysis

Serological and leukocyte infiltration data were analyzed using One Way ANOVA with Tukey’s test and the Mann-Whitney test, respectively. All the statistical analyses were performed using GraphPad Prism 6 software.

## 3. Results

### 3.1. Construction of the Recombinant WNV gE and NS1 Proteins

DNAs encoding the relevant fragments of WNV gE and NS1, respectively, were commercially synthesized (Figure 1) and introduced into the polyhedrine locus of recombinant baculovirus genome bacmids by using Invitrogen’s Baculovirus Gateway technology, as described in Materials and Methods. Proteins expressed from the resulting baculoviruses were to be used as immunizing antigens. The results are shown in Appendix A.

Moreover, shorter fragments of the gE and NS1 proteins were synthesized to serve as templates for ELISA antigens. In the case of gE, domain III was fused to an N-terminal signal sequence and C-terminally complemented with a c-myc tag and a GST fragment. In the case of NS1, a part of the C-terminus was deleted before fusion to c-myc and GST. Recombinant baculovirus bacmids were generated as above (Appendix A). The sequences and orientations of all four bacmid inserts were confirmed by sequencing.

### 3.2. Susceptibility of C57BL/6J Mice to WNV ITA09 Infection and Immunogenicity of WNV Antigens

Since TLR3KO mice are bred on a 129/C57BL/6 background, we characterized the dose-dependent outcome of WNV infection in C57BL/6 mice. As the inoculation of C57BL/6 mice with 10^5^ pfu of our WNV ITA09 stocks led to a consistent high-dose infection of the brain, followed by 100% lethality (approximately 100 LD_50_ (lethal dose for 50% of the animals in the group), we decided to use this dose for the upcoming vaccination-challenge experiments (Appendix A). Moreover, the immunogenicity of our baculovirus-derived WNV antigens was assessed in C57BL/6 mice. The results showed that it took two to three vaccinations with 5 µg of our immunizing antigens to induce a reliable immune response in the immunized mice and, in the case of gE, to induce a reliable response of neutralizing antibodies against WNV (Appendix A).

### 3.3. Immune Protection of Wild Type and TLR3KO Mice Following NS1- and/or gE-Vaccination

To assess the protective effects of NS1- and gE-vaccination against WNV in wild type or TLR3KO animals, 10 groups of mice (Table 1) were immunized three times according to the scheme provided in Appendix A. In the following section, the results are subdivided into serological responses against WNV proteins prior to the challenge and various pathogenic outcomes following inoculating the mice with WNV.

#### 3.3.1. Serologic Responses Prior to Challenge

• Antibodies against NS1

On day 50 post-vaccination (pre-challenge), the sera from mock-immunized mice showed very low reactions against the NS1-ELISA antigen, thus determining the background response to the NS1 ELISA antigen. The sera from WT mice immunized either with NS1 alone, an NS1 and gE combination, or the commercial WNV vaccine showed a significantly elevated response, suggesting that all these individuals had developed an antibody response against NS1 (Figure 2A). Interestingly, the seroreactions of TLR3KO mice immunized with the NS1 antigen did not significantly differ from the reactions of the mock-immunized mice. Thus, the immune response against NS1 of the TLR3KO mice remained uncertain. However, TLR3KO mice immunized with the mixture of gE and NS1 or with the commercial WNV vaccine (Figure 2B) showed seroreactions in the same range as WT mice, suggesting that they had developed antibodies against NS1.

• Antibodies against gE

Additionally, at day 50, the sera from mock-immunized mice showed barely any reaction against the gE-ELISA antigen, thus defining the very low level of negative reactions. Among the animals immunized with gE alone, some individuals remained within this negative range. However, two out of five individuals from the TLR3KO group (Figure 2D) and one out of five individuals from the WT group (Figure 2C) showed clear seroreactions against the gE-ELISA antigen. Moreover, all individuals immunized with the combination of gE and NS1 as well as those immunized with the commercial vaccine showed very clear seroreactions against the gE-ELISA antigen. Thus, against expectations, not all gE-only immunized animals had developed a measurable antibody response against gE by day 50. However, as soon as NS1 became part of the antigenic mixture, very clear antibody responses could be measured.

#### 3.3.2. Neutralizing Antibodies

A similar picture emerged upon performing the PRNT50 assay with the same sera: each two out of five individuals immunized with gE alone (both WT and TLR3KO mice) showed neutralizing activity against WNV. While NS1-immunized or mock-immunized individuals remained seronegative, three out of three individuals from both the WT and TLR3KO groups showed high WNV-neutralizing activity following application of the commercial vaccine. An interesting difference emerged among the animals immunized with the combination of gE and NS1: three out of five WT animals developed neutralizing antibodies (Figure 2E), whereas five out of five TLR3KO individuals that did the same (Figure 2F). Thus, it seemed as if the development of neutralizing antibodies against WNV was facilitated in the presence of the two antigens gE and NS1. Moreover, under these circumstances, the TLR3KO mice developed a more consistent antibody response than WT animals.

### 3.4. Protection against WNV Infection and Disease in WT and TLR3KO Mice Following Immunization

The mice were infected with 10^5^ pfu of WNV ITA09 post-immunization to assess the protective efficacy of the different vaccinations and the pathogenic nature of the infection. The readouts are subdivided below into survival, clinical scores, viral spread, and inflammation in the CNS.

#### 3.4.1. Survival

At 3 weeks post challenge, all the members of the groups immunized with either gE or gE + NS1 or commercial vaccine were still alive, including all those individuals who had not developed a detectable amount of antibodies against gE. However, in the case of NS1 immunization without gE, only two of five and three of five, respectively, of WT and TLR3KO mice survived (Figure 3A,B). Within the mock-immunized control groups, three of five mice in the WT group showed severe WNV infection, whereas all the mice in TLR3KO group succumbed to severe WNV infection and did not survive. Thus, while there was no difference concerning case fatality between mock-immunized and NS1-immunized WT mice, the number of TLR3KO mice succumbing to lethal WNV infection was reduced in NS1-immunized mice compared to mock-immunized individuals.

#### 3.4.2. Clinical Scores

Mice were scored during the entire period of the experiment for clinical signs according to the criteria described in Table 2. Higher scores represent more severe clinical illness, whereas low scores are associated with mild courses.

As shown in Figure 4, high clinical scores were observed in the mock-immunized groups as well as in the groups immunized with NS1/adjuvant alone, which relates well with the survival rate shown previously. Low clinical scores were observed among the groups immunized with gE/adjuvant alone. Although most members of these groups had not developed detectable antibody responses against gE, not one single individual developed severe signs of WNV-disease. Importantly, none of the animals immunized with adjuvant and the combination of glycoprotein E and NS1 reached a clinical score.

Thus, animals immunized with the combination of gE and NS1 were clinically protected. Animals immunized with gE alone, even those without detectable antibodies against gE, were substantially protected from the disease, whereas NS1-immunized animals were hardly better protected than mock-immunized animals. Regardless of the vaccine type, we observed no significant difference between the clinical scores of WT and TLR3KO mice, respectively.

#### 3.4.3. Viral Spread

Viremia is a clear sign of spread of the virus in the host. Therefore, plasma samples were collected on days 2 and 4 post infection and analyzed for WNV RNA by quantitative real time RT-PCR (RT-qPCR). The data of all animals with detectable viremia are presented in Figure 5A,B. Viremia was indeed detected in all mock-immunized animals and in animals immunized with NS1 alone. As soon as gE was included in the vaccine, viremia was reduced, in some cases to undetectable levels. Interestingly, this was indeed true for all gE-immunized animals, independently of their gE antibody titers. Compared to day two, viremia was reduced on day 4 post infection. None of the animals immunized with a gE-component was still viremic at this later time point.

It is understood that WNV reaches the internal organs following viremia [50]. Therefore, we also analyzed the spleens and the brains of all WNV-inoculated animals for WNV RNA by RT-qPCR. The results are summarized in Table 3.

Indeed, WNV-RNA was detected in the brains and spleens of most mock- or NS1-immunized animals, independently of their genetic background. In WT mice, WNV-RNA was detected in the spleens of all animals immunized with NS1 but only in three of five mock-immunized ones. Interestingly, in the TLR3KO group, WNV-RNA in spleens was detected only in three of five mice immunized with NS1 but in all mock-immunized mice (Figure 5C,D).

The brain seemed to be more sensitive for WNV than the spleen since WNV RNA was detected in the brains of all 10 mock immunized animals (five WT and five TLR3KO) as well as in nine of the ten NS1 immunized individuals (five of five WT and four of five TLR3KO). Moreover, WNV RNA in brains was detected in two out of five WT mice immunized with gE alone and in one out of five WT mice immunized with a combination of gE and NS1 (Figure 5E,F). Notably, these three animals had previously been identified as early viremic (Figure 5A).

By contrast, WNV-RNA was neither detected in the brain or spleen of any of the animals vaccinated with the commercial vaccine (three WT and three TLR3KO) nor in any of the TLR3KO mice immunized either with gE alone or with a combination of gE and NS1.

Thus, in the absence of a vaccine containing gE, WNV invariably reached the brains of the inoculated animals. Moreover, in three out of four animals with detectable viremia at day 2 post WNV inoculation, the virus was later on also detected in the brain. Thus, early viremia was a good indicator for WNV brain invasion.

#### 3.4.4. CNS Inflammation

We subjected brain samples of each individual to histological examination. In addition, we quantified the CNS-infiltrating neutrophils, lymphocytes, and monocytes in the infected animals by flow cytometry. Finally, IFNγ and TNFα expression in the brain was determined by RT-qPCR (Appendix A).

Infiltrating cells: hematoxylin and eosin (H&E)-stained brain sections from all mice were screened for perivascular infiltration of leukocytes as well as for necrotic and/or apoptotic areas indicative of CNS inflammation. Exemplary pictures for each scoring level are shown in Appendix A.

The results are shown in Figure 6A–B, where the animals are also stratified into those with detectable WNV RNA in their brains and those without. All mice with detectable WNV RNA in their brains, except one TLR3KO individual (H4, which had been immunized against NS1), received a histological score of one or higher, whereas many mice without detectable WNV had a score of zero. However, there was a great amount of overlap in the scores of one and two, suggesting that histology alone was not a good indicator to discriminate between the two strata.

To refine these crude assessments, flow cytometric analysis (FACS) was used to quantify the infiltrating immune cells in the brain of the infected mice. Briefly, neutrophils (CD11b^+^Ly6G^+^), monocytes (CD11b^+^Ly6C^high^), and lymphocytes (CD45^+^CD11b^-^) were determined (Appendix A). Representative FACS plots and the gating strategy for each population are shown in a supplementary figure (Appendix A. The results of this analysis are summarized in Figure 6C–H.

Animals with detectable WNV RNA in their brains had significantly higher numbers of each single infiltrating cell category than animals without WNV in their brains. However, there was still significant overlap, particularly in the case of the neutrophils and monocytes.

Upon ranking of the data according to neutrophil counts, the lowest values among all mice were observed with C2, a WT mouse that had been immunized against NS1 and had a significant amount of WNV RNA in its brain. On the opposite side, the highest neutrophil counts were observed with K2, a TLR3KO mouse, which had also the highest burden of WNV in its brain.

Upon ranking according to monocytes, the bottom eight positions with the lowest counts were occupied by animals without WNV in their brains, which had been immunized against gE, either alone or in combination with NS1. Both WT and TLR3KO mice were among them. Interestingly, the five top positions with the highest monocyte counts were all occupied by mock-immunized TLR3KO mice (group K) with variable amounts of WNV in their brains. These two extremes were separated by an intermixed group of animals with or without WNV in their brains. However, the number of animals with WNV in their brains generally increased with increasing monocyte counts.

Upon ranking for lymphocyte counts, the 24 bottom positions with the lowest cell counts were all occupied by animals without WNV in their brains, whereas the 18 top positions were occupied by animals with detectable WNV in their brains. Notably, only one single animal (H2) without detectable WNV in its brain ranked within the range of the WNV-positive animals. Again, the five mock-immunized TLR3KO mice (group K) ranked in five of the six top positions.

Notably, in the absence of WNV, the inflammatory cell counts were similar in the brains of TLR3KO mice or their WT counterparts. However, in the presence of comparable WNV loads in their brains (Figure 7A), the total infiltrating inflammatory cell count was highest in the mock-immunized TLR3KO group (Figure 7B). More specifically, the monocyte counts in the mock-immunized TLR3KO mice exceeded the corresponding counts of the other three groups by far (NS1-immunized TLR3KO mice as well as NS1- or mock-immunized WT mice) (Figure 7C). Together, these data suggested that an intact TLR3-pathway functioned to dampen the inflammatory cell invasion upon WNV invasion of the brain and that WNV NS1 contributed to neural inflammation with a mechanism different from repression of TLR3 signaling.

Cytokines represent another and more refined type of inflammatory indicator. IFNγ and TNFα were selected as representative cytokines to address this issue. Therefore, all brain samples were examined for the expression of IFNγ and TNFα, respectively, using RT-qPCR. To make the results comparable among individuals and groups, beta-actin mRNA was used as a reference as described in Materials and Methods. The results are shown in Figure 8, where the animals were also stratified into those with detectable WNV RNA in their brains and those without. Indeed, all animals with detectable WNV RNA in their brains showed also a 10- to 100-fold higher expression of interferon gamma and TNFα mRNA compared to any animal without detectable WNV RNA. These observations confirmed that WNV brain invasion was consistently associated with CNS inflammation. In the case of TNFα expression, high expression levels had an obvious tendency to correlate with high WNV loads in the brain [51].

In order to understand if there is a correlation between the viral load in the brain and cytokine and cellular infiltration, a graph plot was made between the viral load in the brain and TNFα expression or lymphocyte count. The level of TNFα expression correlated relatively well with the WNV titer in the brain, with an R^2^ value of 0.74 (Figure 9A). By contrast, the inflammatory cell counts in the WNV-positive brains seemed not to be strongly influenced by viral loads. As a representative example, the correlation of lymphocytes with WNV titers is shown (Figure 9B), with this correlation having an R^2^ of 0.04. Although the difference was not statistically significant, TNFα and IFNγ levels were higher in WNV-infected TLR3KO mice than WNV-infected WT mice.

## 4. Discussion

After establishing the production of ELISA- and immunizing-WNV antigens in the baculovirus system (S1 through S3), we performed protective studies against WNV brain invasion in WT as well as TLR3KO mice and studied the influences of the two mouse genotypes as well as prior immunization against gE and NS1 towards the severity of brain inflammation. The salient features of our research are as follows:

Despite some suboptimal antibody responses, immunization with a gE antigen provided a good measure of protection against WNV brain invasion, while an additional benefit due to simultaneously vaccinating against NS1 could not readily be detected, which was in agreement with previous studies [13,18,23,27,52]. Indeed, WNV-RNA was detected in the brain of most mock- or solely NS1-immunized animals, independent of their genetic background. Thus, in contrast to others, we were unable to confirm the importance of either prior immunization against NS1 [16,42] or the TLR3 receptor as a mediator for WNV entry to the brain [39]. Similarly, the viral loads in the brains of WT and TLR3KO mice, respectively, did not significantly differ between the two genotypes, which was true for both groups of mice (mock-vaccinated and NS1-vaccinated). A major limitation of our study is the small number of animals used in experiments, which might have been insufficient to detect subtle but significant differences, especially if considering the experimental variation and disease duration, which may occur in mice inoculated with WNV [53]. Although further investigation in larger experimental groups is required to support the results of our study, a comprehensive analysis of factors contributing to encephalitis allowed us to identify a role for NS1 in enhancing the infiltration of immune-inflammatory cells in the WNV-infected brain of TLR3-deficient mice.

Even if high WNV RNA loads were detected in the brains of both wild type and TLR3KO mice (Figure 5E,F and Figure 7A), the mock-vaccinated TLR3KO mice showed the highest degree of inflammatory cell invasion to the brain as a consequence of WNV neuroinvasion (Figure 7B). This observation suggested that the intact TLR3 pathway in WT mice contributed to dampening of the brain inflammation as a consequence of WNV invasion, conceivably through activation of downstream effectors and the interferon response [54]. This observation is in agreement with a previous report, showing that the absence of the TLR3 receptor led to increased inflammation in the brains of WNV-infected mice as a consequence of increased viral loads [38,55].

The most interesting observations in our study were made with NS1-immunized mice. In fact, NS1-immunized TLR3KO mice exhibited less inflammatory cell infiltration (mainly monocytes, Figure 7C) upon WNV infection than mock-immunized TLR3KO mice, at viral loads comparable with that detected in control WT mice. Further studies are needed to clarify the mechanisms of protection of TLR3-deficient mice against WNV neuroinvasion and brain inflammation through immunization with NS1. Based on the results of our study, we hypothesize that TLR3 deficiency led to an impaired innate immune response and hence to unrestricted viral infection and replication in neurons, brain inflammation with infiltration of inflammatory CD11b^+^Ly6C^high^ monocytes, production of proinflammatory cytokines, and increased permeability of the blood–brain barrier, as also described in TLR3-deficient mice infected with Japanese encephalitis virus, a neurotropic flavivirus that is very similar to WNV [56]. WNV NS1, besides inhibiting the TLR3 pathway, has a direct effect on the permeability of the blood–brain barrier, and this effect could be synergistic in mice with TLR3 deficiency. Immunization against NS1 could prevent damage to the blood–brain barrier, with mechanisms that are independent from TLR3, but are relevant in the context of an impaired innate immunity. In contrast to others, we have not observed adverse effects upon immunization with NS1 [37]. While further studies using various WNV strains in appropriate animal models, particularly non-human primates, are warranted to clarify the interplay between WNV NS1 and the host TLR3-dependent innate immune response, our findings highlight the beneficial effect of immunization with NS1 in reducing neuroinflammation upon WNV infection in individuals with impaired innate immunity.

## 5. Conclusions

In our experiments, the degree of inflammation measured in the brain was affected by viral load, TLR3 signaling, and previous immunization again NS1. Viral load in the brain had a dose-dependent effect on TNF alpha expression in both WT and TLR3KO mice, whereas the degree of cellular infiltration in WNV-infected brains was less severe in mice with an intact TLR3 pathway than in those with TLR3 deficiency. At variance, TLR3KO mice immunized against WNV NS1 showed reduced inflammatory cell infiltration, which was comparable to the degree observed in WT mice. Apart from the fact that gE and NS1 seemed to adjuvant each other, inclusion of NS1 into WNV vaccines may be considered in order to extend protection against neuroinvasive WNV to individuals with genetic or functional defects of innate antiviral immunity. With respect to potential animal species-specific adverse effects attributed by others to immunization with NS1 [37], it would be crucial to address this issue in non-human primate models.

## Figures and Tables

**Figure 1 viruses-11-00603-f001:**
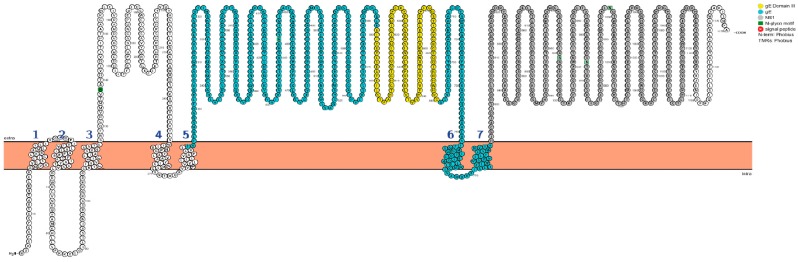
Schematic representation of the West Nile Virus (WNV) N-terminal polyprotein with numbered amino acids (one-letter codes; beginning from the N-terminal (H_2_N) methionine (M)) and predicted transmembrane domains (bold numbers 1–7). The gE sequence is presented in blue with its neutralization domain III in yellow. NS1 is depicted in grey. External parts of the sequence (labeled extra) are depicted above the symbolic membrane and cytoplasmic parts are shown below (labeled intra). For producing immunizing antigens, the external domains of gE and NS1, respectively, were bracketed each with a synthetic N-terminal signal sequence and two C-terminal tag epitopes (V5 and 6× his). For the ELISA antigens, the DIII domain of gE was supplemented with the synthetic N-terminal signal sequence and with a C-terminal c-myc-GST-tail. Similarly, the NS1 fragment from aa 792 to 1050 was bracketed with the same features. (Picture exported from PROTTER, Protter: interactive protein feature visualization and integration with experimental proteomic data) [49].

**Figure 2 viruses-11-00603-f002:**
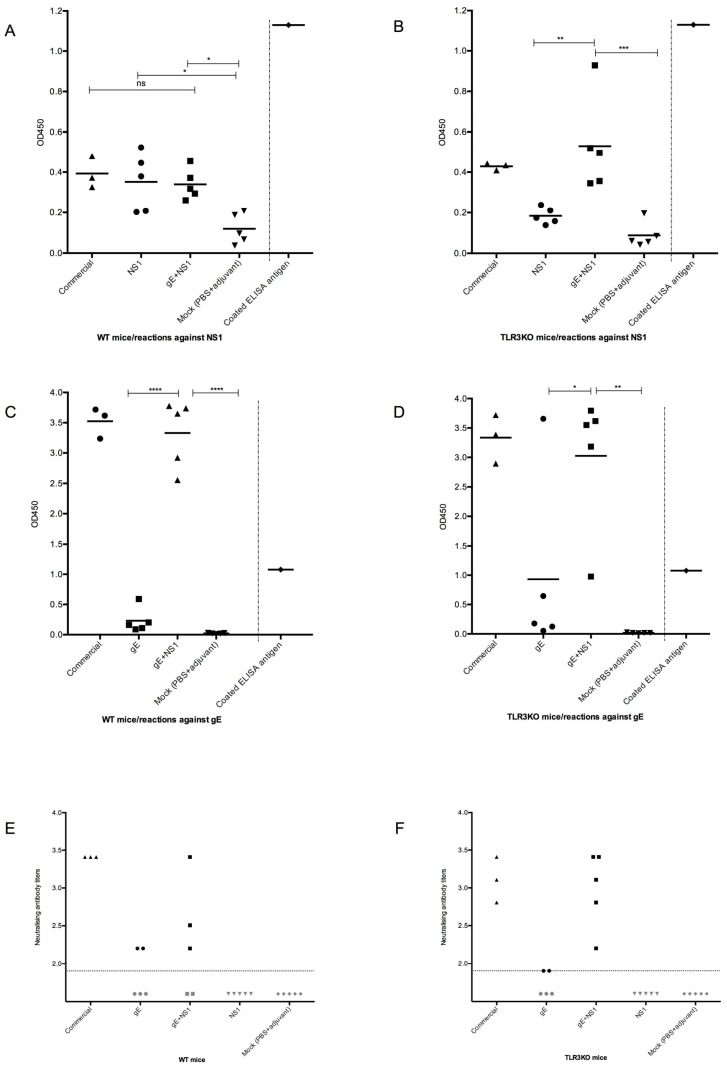
Antibody responses prior to WNV challenge infection. The groups of mice tested (see Table 1) are indicated on the x-axis. Panels (**A**), (**C**), and (**E**) represent WT mice; panels (**B**), (**D**), and (**F**) represent TLR3KO mice. Each symbol represents an individual mouse; the horizontal line indicates the mean value for each group. Panels (**A**–**D**) show ELISA values as measured at OD450 nm (y-axis). (**A**,**B**): ELISA for NS1 antibodies; (**C**,**D**): ELISA for gE antibodies. All sera were tested on the same plate. Separated by a dotted line on the right of each panel (**A**–**D**) the value of a positive control is given, which was determined by using an anti-C-myc monoclonal antibody for testing the amount of coated antigen. Panels (**E**,**F**): WNV-neutralizing antibodies as detected by plaque reduction neutralization assay (PRNT), with the titer indicated on the y-axis. The horizontal dotted line indicates the negative cut-off value, which equals a serum dilution of 1:80, the lowest dilution used in the assay. The values below this cut-off are fictional. Where applicable, one-way ANOVA with Tukey’s test for multiple comparisons was used to investigate statistical differences between the groups. Significant differences are indicated by asterisks.

**Figure 3 viruses-11-00603-f003:**
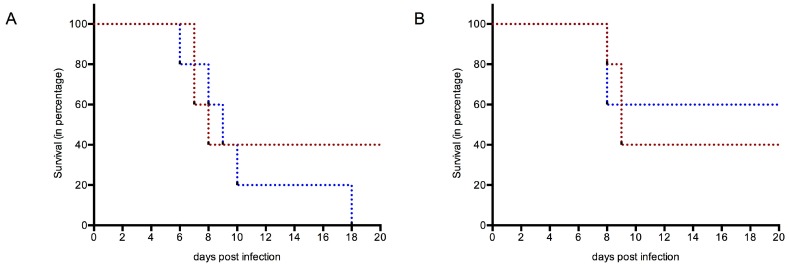
Survival of mice after experimental WNV infection. (**A**) Mock immunized; (**B**) NS1 immunized. Red dotted lines used to show WT mice; blue dotted lines used to indicate TLR3KO mice. The time in days is given on the x-axis, starting at the day of experimental inoculation with 100 LD_50_ of WNV Italy 09. The y-axis gives the percentage of surviving animals (for additional information see Table 1). Note: In order to not overload the figure, the data of groups with a 100% survival rate were not included in this figure.

**Figure 4 viruses-11-00603-f004:**
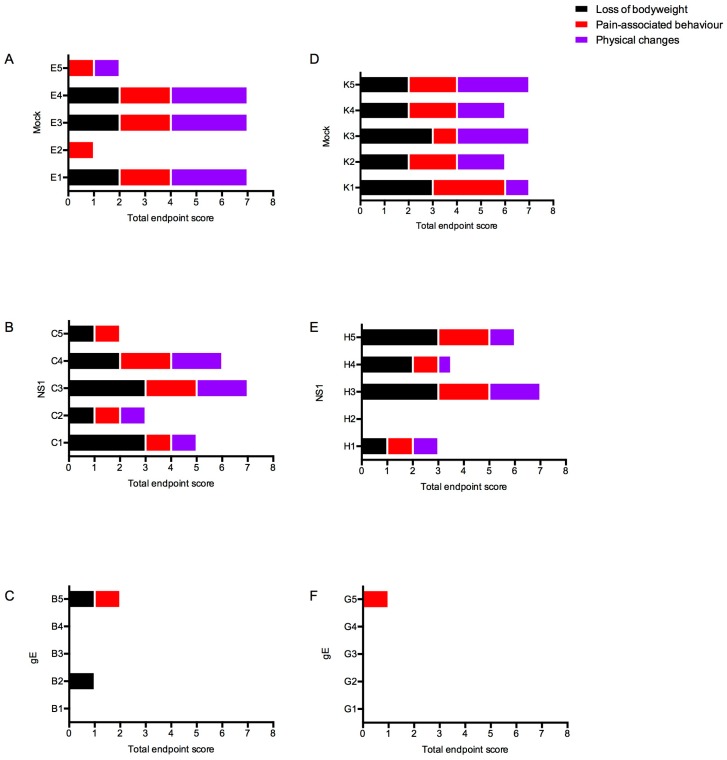
Clinical scores post challenge with WNV. Panels (**A**), (**B**), and (**C**) show data for wild type mice; (**D**), (**E**), and (**F**) show data for TLR3KO mice. (**A**,**D**): mock-immunized animals; (**B**,**E**): NS1-immunized animals; (**C**,**F**): gE-immunized animals. Each bar represents the cumulative clinical score of one individual mouse at its day of euthanasia. The y-axis represents the individual mice per immunized group having a total score of 1 or more. Mice counted as survivors were euthanized at day 19 post challenge. The bars are color-coded as follows: black represents points allocated for weight loss; red represents pain-associated behavior; purple represents physical changes. Note: In order to not overload the figure with empty graphs, the data of mice immunized with both gE and NS1, all of which did not show any measurable disease signs, are not included in this figure.

**Figure 5 viruses-11-00603-f005:**
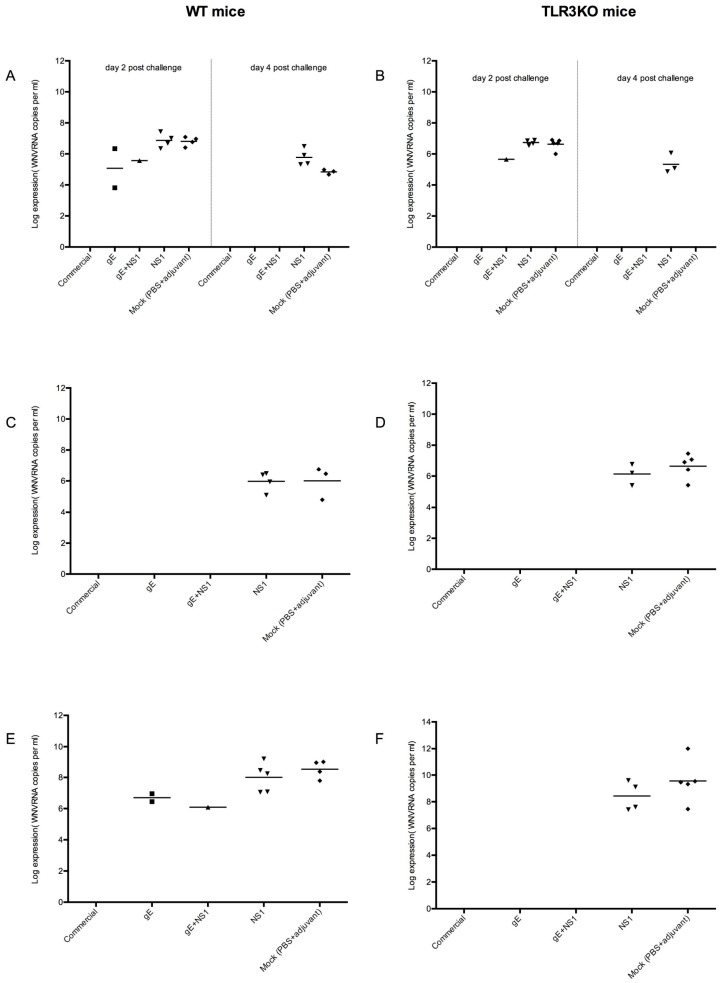
Levels of WNV RNA in the organs of WNV-infected mice. Separate panels are shown for plasma (**A** and **B** are each split into day 2 on the left and day 4 on the right), spleen (**C**,**D**), and brain (**E**,**F**) samples. Panels (**A**,**C**,**E**): WT mice; (**B**,**D**,**F**): TLR3KO mice. The loads of WNV in the individual samples were determined by RT-qPCR and each dot in each panel represents one single animal. The y-axis of each panel represents the logarithmic value of viral genome copies detected per mL of either blood plasma or organ suspension. The individual groups of animals are listed on the x-axis. Note: Samples which did not provide a Ct value, i.e. where WNV could not be detected, have been excluded from the figure.

**Figure 6 viruses-11-00603-f006:**
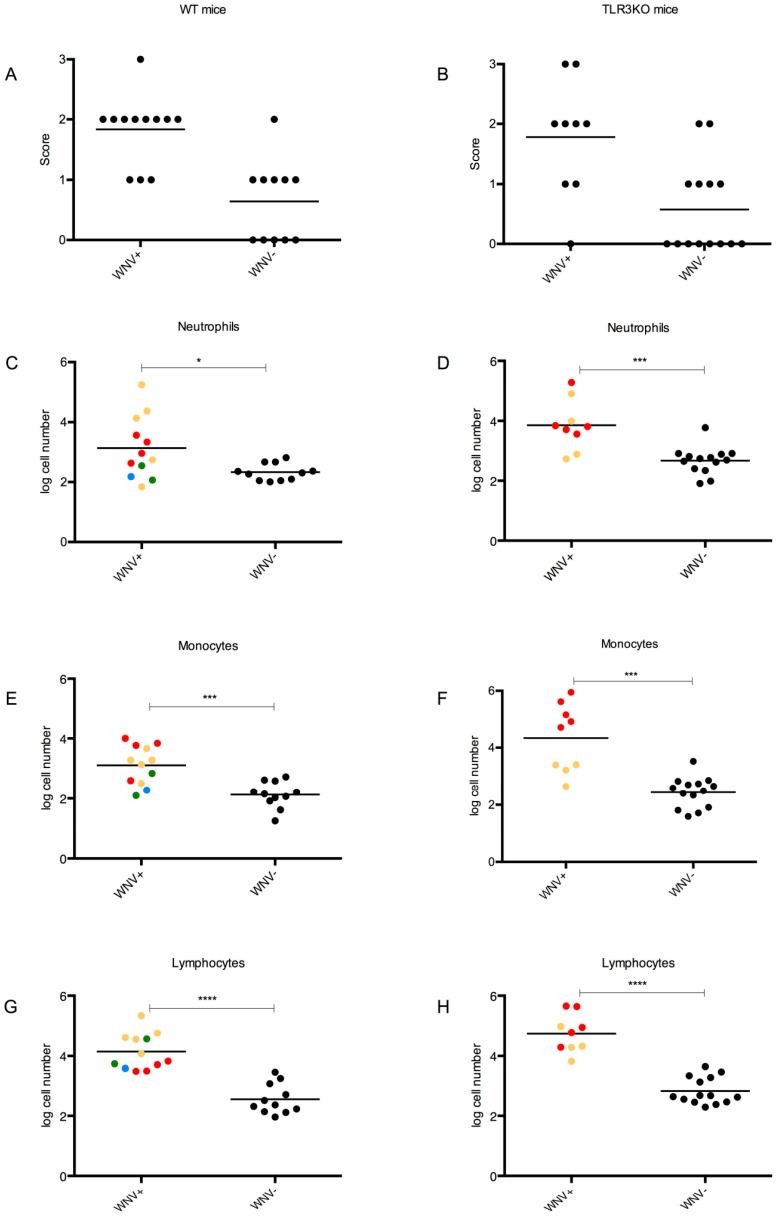
Leukocyte infiltration upon WNV infection of the brain. The brains of all experimental wild type (**A**,**C**,**E**,**G**) and TLR3KO mice (**B**,**D**,**F**,**H**) were subjected to histological scoring (**A**,**B**) and flow cytometric analysis (FACS) for the detection and quantification of inflammatory cells, including neutrophils (**C**,**D**), monocytes (**E**,**F**), and lymphocytes (**G**,**D**). To recognize the role of WNV in the context of CNS inflammation, the individuals were stratified into those with detectable WNV RNA in their brains (WNV^+^) and those without (WNV^−^). WNV^+^ individuals have been further color coded to represent different groups: gE green, NS1 yellow, gE + NS1 blue, and mock red, respectively. In addition to the individual values, the mean is also presented. The Mann-Whitney test was used to investigate statistical differences between the groups. Significant differences are indicated by asterisks with *, *p* < 0.05; ***, *p* < 0.001; ****, *p* < 0.0001.

**Figure 7 viruses-11-00603-f007:**
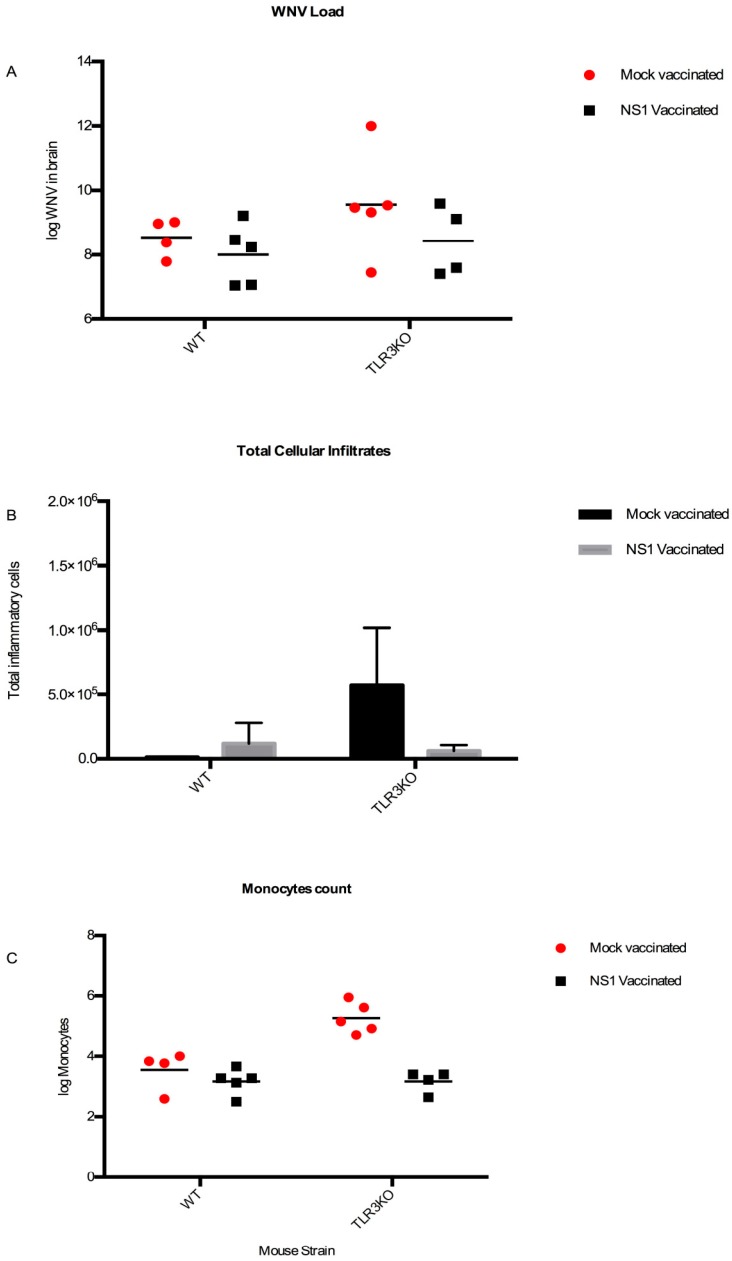
WNV loads and total cellular and monocytic infiltrates after NS1- or mock-vaccination. (**A**) log WNV viral copies per mL (y-axis) in the brains of the WNV-positive NS1- and mock-immunized mice, respectively. (**B**) The number of granulocytes, monocytes, and lymphocytes were added to each other to give the total amount of infiltrating inflammatory cells in the brains (y-axis) of the same mice. (**C**) log monocyte counts in the brains (y-axis) of WNV-positive NS1- (black) and mock- (red) immunized mice, respectively.

**Figure 8 viruses-11-00603-f008:**
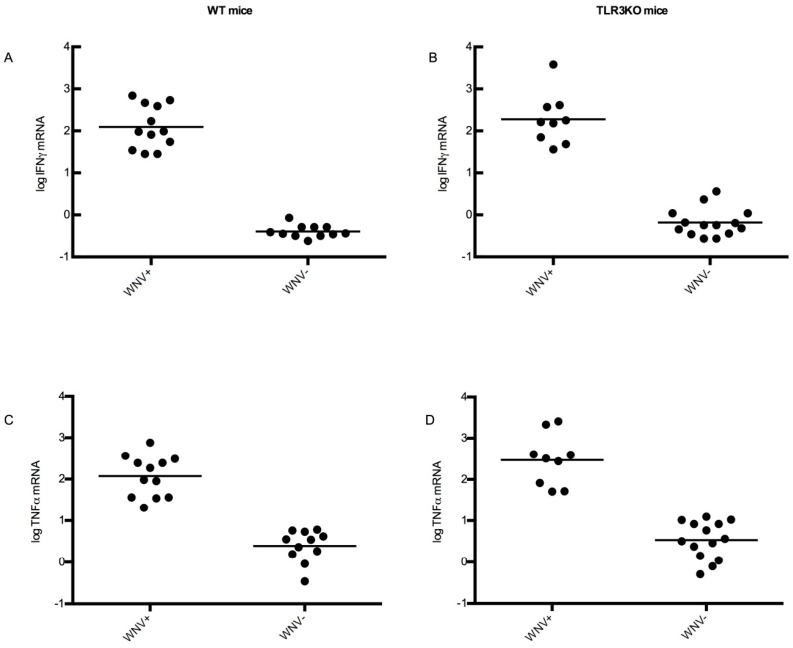
Cytokine response upon WNV infection of the brain. RNA was extracted from brains of all experimental mice and analyzed for cytokine RNAs by qRT-PCR. The Ct values were set in relation to beta-actin mRNA as described in Materials and Methods. On the y-axis, the results of these calculations are shown as log10 values. The data were stratified into a group representing animals with WNV RNA in their brain tissue (WNV^+^) and those without (WNV^−^). Panel (**A**): IFNγ mRNA in WT mice. Panel (**B**): the same in TLR3KO mice. Panel (**C**): TNFα mRNA in WT mice. Panel (**D**): the same in TLR3KO mice. All individual values as well as the mean are indicated.

**Figure 9 viruses-11-00603-f009:**
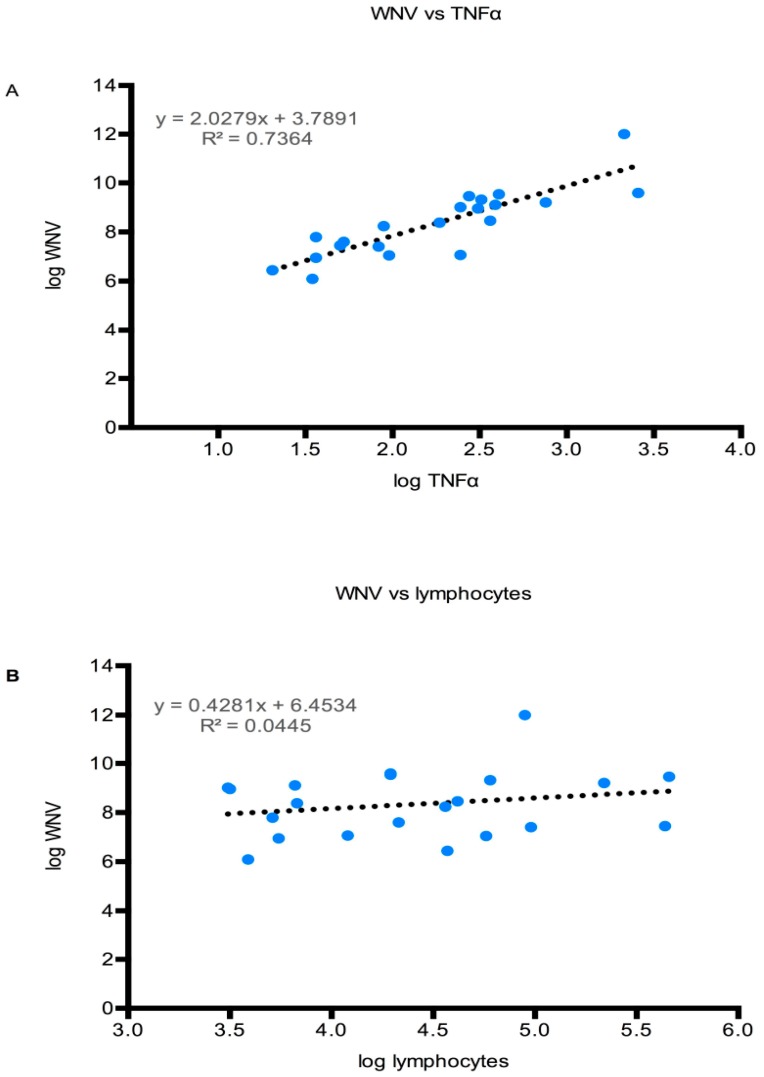
Viral load in the CNS correlates largely with TNFα expression but not with lymphocyte counts. The WNV load in the brains of both WT and TLR3KO mice were expressed as log10 values and paired with (**A**) log10 values of either TNFα mRNA or (**B**) with log10 values of lymphocyte counts in the CNS of each individual before being presented in an X,Y graph. The regression formula as well as the R^2^ values are indicated in each graph.

**Table 1 viruses-11-00603-t001:** Overview of group designations, mouse genotypes, and vaccinating antigens.

Group	#	Genotype	Vaccination
**A**	3	^a^WT C57BL/6	Commercial
**B**	5	WT C57BL/6	^c^gE
**C**	5	WT C57BL/6	^d^NS1
**D**	5	WT C57BL/6	gE + NS1
**E**	5	WT C57BL/6	Mock (^e^PBS + adjuvant)
**F**	3	^b^TLR3KO	Commercial
**G**	5	TLR3KO	gE
**H**	5	TLR3KO	NS1
**J**	5	TLR3KO	gE + NS1
**K**	5	TLR3KO	Mock (PBS + adjuvant)

^a^WT: wild type; ^b^TLR3KO: Toll-like receptor 3 knockout; ^c^gE: envelope glycoprotein; ^d^NS1: non-structural protein 1; ^e^PBS: phosphate-buffered saline.

**Table 2 viruses-11-00603-t002:** Clinical scoring.

Symptom	Observation	Score
Loss of body weight	Body weight within range of untreated group	0
	1–5% loss of body weight	1
	>5–15% loss	2
	>15% loss	3
Pain-associated behavior	Normal activity	0
	Reduced activity, moving after slight impulse, shivering	1
	Inactive, apathic, moving after moderate impulse	2
	Ataxia, paralysis, not moving even after moderate stimulation	3
	Normal posture	0
	Hunched posture	1
	Hunched posture, head lying on the cage floor	2
	Lying prone on the cage floor	3
Physical changes	Normal hair coat	0
	Ruffled	1
	Ruffled, loss of fur, ungroomed	2
	Normal eyes	0
	Eyes closed or squinted, no discharge	1
	Eyes closed or squinted, with discharge	2

**Table 3 viruses-11-00603-t003:** WNV RNA in brain and spleen following immunization and WNV challenge infection.

Immunization	WT Mice		TLR3KO	
	Brain ^a^	Spleen ^b^	Brain	Spleen
gE	2/5 ^c^	0/5	0/5	0/5
NS1	5/5	4/5	4/5	3/5
gE + NS1	1/5	0/5	0/5	0/5
Commercial vaccine	0/3	0/3	0/3	0/3
PBS + adjuvant	4/5	3/5	5/5	5/5

^a,b^ Detection of WNV RNA in organ extracts by quantitative real time RT-PCR (RT-qPCR). ^c^ Number of positive animals/number of tested animals.

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
