# Peer review of "Role of NS1 and TLR3 in Pathogenesis and Immunity of WNV"

_viruses, 2019, doi:10.3390/v11070603_

Round 1
Reviewer 1 Report
The paper by Patel et al. is a well written manuscript that describes the role of NS1 and TLR3 in pathogenesis and immunity after immunization with WNV recombinant proteins. The paper is clearly structured and the study was carried out carefully.This reviewer agrees with the general design of the study and with the methods used. Only some minor corrections are necessary.
The following minor revisions should be done:
- under 2.4 Animal and protection study: Inoculation route of mice with WNV should be mentioned.
- Overall in the manuscript: Check spaces between number and unit
- Overall in the manuscript: Make sure that "µ" is used instead of "u" in Units
Line 235: Cell number is probably 5x105 not 5x105
Line 237: "CO2" should be written as CO2
Line 261: "Sybr" should be written as "SYBR"
Line 268: "DDCt" should be written as "ΔΔCt"
Line 332: Closing bracket at end of paragraph is missing
Fig.2: Scaling of the y-axe: negative values are not needed in A - D
Fig. 4: Labelling of graphs is missing (A - E)
Line 607 and 615: Format of citations should be checked
Supplemental Material: Fig S2: Part B is not readable. Labelling A) and B9 should be same size. Several times in Supplemental material “ELISA antigen production”: Centrifugation speed is written as 3000g, 3000xg, 3000x g….
Author Response
Thank you, we appreciate your efforts.
Question | Reviewer 1 comment | Our answer |
1. | under 2.4 Animal and protection study: Inoculation route of mice with WNV should be mentioned. | Corrected (i.p.) |
2. | Overall in the manuscript: Check spaces between number and unit | Thank you. Corrected |
3. | Overall in the manuscript: Make sure that "µ" is used instead of "u" in Units | Corrected |
4. | Line 235: Cell number is probably 5x105 not 5x105 | Thank you. Corrected |
5. | Line 237: "CO2" should be written as CO2 | Corrected |
6. | Line 261: "Sybr" should be written as "SYBR" | Corrected |
7. | Line 268: "DDCt" should be written as "ΔΔCt” | Corrected |
8. | Line 332: Closing bracket at end of paragraph is missing | Corrected |
9. | Fig.2: Scaling of the y-axe: negative values are not needed in A – D | Corrected |
10. | Fig. 4: Labelling of graphs is missing (A - E) | Corrected |
11. | Line 607 and 615: Format of citations should be checked | Corrected |
12. | Supplemental Material: Fig S2: Part B is not readable. Labelling A) and B9 should be same size. Several times in Supplemental material “ELISA antigen production”: Centrifugation speed is written as 3000g, 3000xg, 3000x g…. | Corrected |

Reviewer 2 Report
Overall, this research is very interesting. Confirms that TLR3 plays a role in brain cell infiltrates during WNV infection. However, the clinical relevance of this should be discussed in more detail. If NS1 vaccination compensates for a lack of TLR3, how does this translate to a useful vaccine for a population? What pathways could be at work that could be targeted in the future? What is the future direction of this research?
The introduction is very long and must be shortened. The paragraphs are repetitive in several ways and could be more concise. Lines 97-100 provide important information, but the thought is not well developed. It is unclear why this is necessary information at this stage of the intro. Lines 102-105 are data unnecessary for the intro.
Methods:
Line 150: “The virus” is unclear which strain you are discussing
Line 152: add city/state/country info for the cells discussed
Line 173: To confirm, 105 pfu was used, not 10^5 pfu?
The thorough description of animal signs of disease is appreciated, perhaps a table would make it easier for the reader to follow.
The methods are generally in sufficient detail for the study to be replicated by another researcher.
Missing a section to describe the statistical analyses performed and software used for each test.
Results:
Overall logical flow of information. Some concerns:
Figure 2: Why do the y-axis have a negative OD value reflected?
Figure 3: It would be helpful as a reader to see these survival curves combined to compare WT and TLR3-/- mice survival. If this makes the graph too busy, try doing smaller panels to reflect the different vaccines used.
Lines 420-422: a table to represent this would be helpful for the reader
Figure 4: The y-axis is not explained, the color key is too small
Section 3.3.3: Several typos are throughout this section. Additional question: what was the justification for using plasma only? WB is known to have detectable virus longer than other blood fluid fractions, did the authors consider evaluating this?
Lines 455-459: What is the significance of this? It appears that, statistically, there is no difference here, and the section is verbose
Lines 462-466: what organs are you discussing here? It is unclear as the explanation progresses.
A table to summarize these results would make it easier to understand. The text is difficult to follow.
Lines 485-489: This is not necessary, present the results without a miniature intro.
Lines 509-512: This seems to depend on the results of one single animal. How relevant is this?
The summary starting on Line 537 is good, but the wording of the entire results section can be confusing.
Lines 554 (and elsewhere): appears that special characters have been removed and are blank.
Figure 9: All other figures were clearly created using a different software. This figure is a lower quality and fuzzy to read. In the text, correct R2 to a superscript 2.
Discussion:
Intro paragraph needs work. As a whole, the writing of this section needs to be more concise.
Line 615: “Neuroinvasion of WNV was the primary cause for various inflammatory signs in the brain”. This is a very obvious statement and is supported by years of other research. Instead, discuss what signals were initiated by neuroinvasion.
The authors need to discuss more about NS1’s role when TLR3 is absent. This is currently presented as an interesting point, but the application is not discussed. Potential pathways could be discussed.
Conclusion:
How does this relate to WNV infections in individuals that don’t have TLR3 deficits? What else do these data tell you?
General comments:
Be consistent with the way numbers are written. Generally, <10 is written in text format, while numbers >10 are numerical in the text. This is not consistently done throughout the manuscript. “Thus” is overused and repeated (ex: Lines 472 and 475).
The paper has interesting and valuable data, but needs more clarity when presenting the information in the text.
Author Response
Thank you, we appreciate your efforts.
Question | Reviewer 2 comment | Answer |
1. | Overall, this research is very interesting. Confirms that TLR3 plays a role in brain cell infiltrates during WNV infection.
| Thank you
|
However, the clinical relevance of this should be discussed in more detail. | We addressed this issue in the revised manuscript; Lines 51- 53 and 596-602. | |
If NS1 vaccination compensates for a lack of TLR3, how does this translate to a useful vaccine for a population? | Addressed in the Discussion and in Conclusions, particularly Line 612. | |
What pathways could be at work that could be targeted in the future? What is the future direction of this research?
| Addressed on Lines 572 onwards and 587 onwards. | |
2. | The introduction is very long and must be shortened. The paragraphs are repetitive in several ways and could be more concise. | We reduced the introduction by approximately 25% according to the suggestions of the reviewer. |
Lines 97-100 provide important information, but the thought is not well developed. It is unclear why this is necessary information at this stage of the intro. | Shortened and reworded more concisely | |
Lines 102-105 are data unnecessary for the intro. | Deleted | |
3. | Line 150: “The virus” is unclear which strain you are discussing | Clarified: WNV ITA09 |
4. | Line 152: add city/state/country info for the cells discussed | Done |
5. | Line 173: To confirm, 105 pfu was used, not 10^5 pfu? | Yes, 10^5. Thank you. Corrected. |
6. | The thorough description of animal signs of disease is appreciated, perhaps a table would make it easier for the reader to follow. | New Table 2, replacing the original text |
7. | Missing a section to describe the statistical analyses performed and software used for each test. | Inserted in Materials and Methods as section 2.13. |
8. | Figure 2: Why do the y-axis have a negative OD value reflected? | Good point. Negative OD values deleted. |
9. | Figure 3: It would be helpful as a reader to see these survival curves combined to compare WT and TLR3-/- mice survival. If this makes the graph too busy, try doing smaller panels to reflect the different vaccines used. | We re-designed Figure 3, making two panels and leaving out all groups with 100% survival (these are mentioned in the text). |
10. | Lines 420-422: a table to represent this would be helpful for the reader | Table 2 inserted. |
11. | Figure 4: The y-axis is not explained, the color key is too small | Figure legend modified to explain y-axis. Color key increased from font size 9 to 12. |
12. | Section 3.3.3: Several typos are throughout this section.
| This is now Section 3.4.3 |
Additional question: what was the justification for using plasma only? WB is known to have detectable virus longer than other blood fluid fractions, did the authors consider evaluating this? | We agree. However, we opted for the cell-free variant in order to detect viral RNA from cell-free virus particles, which most closely reflect infectious virus. | |
13. | Lines 455-459: What is the significance of this? It appears that, statistically, there is no difference here, and the section is verbose | The paragraph about viral spread was strongly edited and supplemented with a new Table 3 |
14. | Lines 462-466: what organs are you discussing here? It is unclear as the explanation progresses | In principle, the blood stream may bring WNV to each single internal organ. For our study, we selected spleen (easy to test but not so relevant for the disease) and brain (most relevant for WNV disease). |
15. | A table to summarize these results would make it easier to understand. The text is difficult to follow. | New Table 3 inserted. |
16. | Lines 485-489: This is not necessary, present the results without a miniature intro. | Deleted |
17. | Lines 509-512: This seems to depend on the results of one single animal. How relevant is this? | As explained in the revised text (Lines 474 onward), we used ranking of the animals according to various parameters as a help to understand the data. Some observations seemed to be noteworthy even without statistical significance. |
18. | The summary starting on Line 537 is good, but the wording of the entire results section can be confusing. | Paragraph reworded to clarify (Lines 502 onward) |
19. | Lines 554 (and elsewhere): appears that special characters have been removed and are blank. | Corrected |
20. | Figure 9: All other figures were clearly created using a different software. This figure is a lower quality and fuzzy to read. In the text, correct R2 to a superscript 2. | Figure 9 revised; superscript corrected |
21. | Intro paragraph needs work. As a whole, the writing of this section needs to be more concise. | Entire discussion revised |
22. | Line 615: “Neuroinvasion of WNV was the primary cause for various inflammatory signs in the brain”. This is a very obvious statement and is supported by years of other research. Instead, discuss what signals were initiated by neuroinvasion. | Entire discussion revised |
23. | The authors need to discuss more about NS1’s role when TLR3 is absent. This is currently presented as an interesting point, but the application is not discussed. Potential pathways could be discussed. | Entire discussion revised. Relevant references included. Potential pathways addressed |
24. | How does this relate to WNV infections in individuals that don’t have TLR3 deficits? What else do these data tell you? | Apparently, immunization against NS1 is less relevant in individuals with an intact TLR3 pathway, yet, gE and NS1 antigens seemed also to adjuvant each other (Figure 2, Table 3, Clinical score, Conclusions) |
25. | Be consistent with the way numbers are written. Generally, <10 is written in text format, while numbers >10 are numerical in the text. This is not consistently done throughout the manuscript. “Thus” is overused and repeated (ex: Lines 472 and 475). | Corrected |
26. | The paper has interesting and valuable data, but needs more clarity when presenting the information in the text. | Done |

Reviewer 3 Report
Please see the document adjoined.
The manuscript describes a thorough work aiming at studying the role of TLR3, with special focus on NS1, in the pathogenesis of WNV infection and immune protection, using a murine model. The work in general is well conceived, well-written and technically sound. Overall it finds very interesting phenomena that merit publication. However, in this revision I have detected some flaws that need to be addressed before it’s ready for publication. A number of them (“major points”) are important gaps that demand a careful response from the authors. The rest (“minor points”) are mostly formal, requiring certain modifications of the text.
Major points:
1. Low number of individuals per group and lack of sound statistics in many of the analyses performed: Five individuals per group may be insufficient if the aim is to detect significant differences between treatments. This problem affects the whole study and can hardly be solved now. I understand the ethical constrains in numbers of animals used in experimental work, but indeed, choosing insufficient number of individuals for an experiment can be unethical if no clear conclusion over the experiment is to be obtained. I do not mean that the work is unreliable. By opposite, I believe that the main conclusions are well supported, mostly due to the thorough experimental work done using a wide range of methodologies intended to better characterize the different mechanisms displayed during WNV infection. But the authors need to bear this gap in mind throughout the text to avoid misinterpretations that are not formally supported statistically. For instance, survival numbers obtained in each group are somehow indicative, but there is no statistically supported difference between groups. Same for seropositive individuals, etc. Under this perspective, the word “difference” needs clarification: e.g. in the results section, paragraph 3.2.2 (“Neutralizing antibodies”), the sentence “An interesting difference…” pointing out that 3 over 5 is different from 5 over 5, the authors should think: “how many times I would expect to get this same difference if I repeat the experiment?” As experimental variation in animal models is usually high (an interesting discussion concerning experimental variation in mice inoculated with WNV can be found in Pérez-Ramírez, E. et al, JGV 2017; 98:662-670) I would suggest the authors to acknowledge this limitation in the text (especially in the discussion section) and indicate that further experimentation (with larger groups) is needed to statistically confirm the observed results.
2. Anti-NS1 antibody determinations. The data obtained using the ELISA developed by the authors to detect anti-NS1 antibodies, which are presented in figure 2 A-B, are very doubtful. I do not mean the ELISA is not trustful. In fact its development is elegant and apparently correct. My concern is over the results and not over the ELISA. One issue is that the results shown in figure S4 c) barely agree with those in fig 2 A-B. In the first, almost all sera reach O.D. > 0.5 and the mean is well above this value, whereas in the second the OD values are <0.5 in almost all cases. Is there any explanation for this disagreement? Indeed, the y-axes scale should be chosen to better observe these differences. I suggest choosing OD scales from 0 to 1. Another issue is that the values obtained are so low that the differences between groups are very small, thus raising a question: do these small differences respond to true differences in NS1-specific antibody levels? In this regard, it is worth to remind that background noise in ELISAs may vary depending on overall serum gammaglobulin levels, which are affected by treatments such as inoculation of immune stimulators. How can the authors ensure that the small differences observed among groups are due to NS1-specific antibodies, and do not respond to different non-specific gammaglobulin levels (or other factors)? Another recommendation for the authors at this respect is to test sera from infected mice (e.g. if obtained, those from preliminary assays to test WNV susceptibility in C57BL76 mice) on this NS1 ELISA, as well as on the DIII-gE ELISAs. These analyses will serve to two different purposes: on the one hand, to validate the ELISAs (a formal validation is lacking in this study) and, on the other hand, to compare the titres (I would strongly recommend titration in parallel) obtained in sera from infected vs immunized mice: by doing this you will have a more accurate idea on how far (or close) to the physiological Ab levels (i.e. the levels attained during a normal immune response to WNV infection) are the NS1 and gE Ab levels measured in the different experimental groups immunized with NS1, gE, NS1 plus gE, and mock. Note that although the group vaccinated with Equip commercial vaccine would do the job for gE-specific antibodies, this is not the case for NS1 Abs, which can hardly be stimulated by a vaccine consisting of an inactivated virus such as Equip. To finish with this point, please bear in mind that NS1 has shown to be a bad immunogen in some animal models (Rebollo, B et al, Comp Immunol Microbiol Infect Dis. 2018; 56:30-33) so there is some basis for expecting a low Ab response to NS1 in NS1-immunized individuals.
3. Effect of the time post-infection disregarded: In the analyses performed in organs the authors do not differentiate the results by the time point (dpi) they were obtained. Obviously, studies in organs arise from individuals either succumbing to the infection (i.e. during a time window of 6 to 10 dpi) or euthanized at the end of the experiment (21 dpi). This is crucial to interpret the results since, for instance, it is relatively rare to find positive viral loads in organs in survivors, while these viral loads are expected to be high in mice succumbing to the infection. This effect explains a lot of the results obtained because surviving the infection is expected to strongly correlate with low viral loads in organs, mild clinical outcomes, low inflammatory responses (cell counts, citokines, etc) in the brain and so on. For instance, in the results section 3.3.3 “Viral spread” (page 14) lines 458-460, the results are described as if the groups were homogeneous (“In the TL3KO group, WNV-RNA was detected in 3 out of 5 mice vaccinated with NS1 and all mock vaccinated mice”). The reader may legitimately wonder if those WNV-RNA positive spleens are from those mice succumbing to the infection, i.e. 3/5 from the NS1 group and 5/5 from the mock group (or otherwise). In any case, the authors should indicate this fact in the text, and probably mention this effect in the discussion. The same applies for other analyses in which the time points are important, i.e. those described under sections 3.3.3 and 3.3.4.
Minor points:
a. In the beginning of the introduction, it should be mentioned that WNV is pathogenic not only for humans and horses but also for a wide range of wild bird species.
b. “Originally from Africa” (page 1 line 38): Better “First known to Africa” since the origin of WNV remains unclear.
c. Id line 43: “…candidate vaccines have been extensively tested…” : please add: “in animal models”.
d. Page 2 line 50: non-structural proteins include NS2a, NS2b, NS4a and NS4b polypeptides. Please include.
e. Id, line 92: For NS1 there is also evidence of negative effects on protection (see Rebollo, B et al. Comp Immunol Microbiol Infect Dis. 2018; 56:30-33, above cited).
f. Throughout the text, the word “vaccinated” is often misused: the only true vaccinated individuals are those receiving the commercial Equip vaccine, while mice receiving doses of recombinant proteins expected to elicit an immune response should be better designated as “immunized” rather than “vaccinated”.
g. Materials and Methods section: page 4 section 2.3 “Virus strains and cells”: Vero cells are mentioned twice, and the full description is not included in the first mention, but in the second. Please correct.
h. Throughout the text the virus dose used in the inoculations reads “105” (one hundred and five) whereas it should read 105 (ten to the five, or one hundred thousands) (similar mistake with H2SO4 instead H2SO4 in page 5). Please correct.
i. Inoculum: at several instances in the text it is stated that the inoculum injected to mice (105 pfu/mouse) is 100x LD50. However, the LD50 calculated for this strain in a previous work (cited in the text: reference #41) was 2.69 TCID50. This value is consistent with that found in another study (Pérez-Ramirez, E. et al, JGV 2017; 98:662-670, above cited) for the closely related strain IT 15803/2008 (LD50= 3.16 pfu). This means that 100 x LD50 is around 3 x102 and not 105 pfu. The authors should explain this discrepancy (perhaps linked to the mouse strain used in the experiment?). Please correct the numbers accordingly if needed.
j. Note that in page 8 the first paragraph (“Since TLR3KO mice…”) has no section heading. It should have been numbered section 3.2, and the following sections renumbered accordingly.
k. Figure 4: Panel marks (A, B, C) are lacking in the figure. Also, in the figure legend, it is unclear whether the clinical scores are cumulative, since it reads: “Each bar represents the clinical score of one individual mouse at its day of euthanasia”. Please correct to state clearly the scores are cumulative.
l. Section 3.3.4 (Page 16): Flow cytometry results are presented in a most confusing way: Lines 509-512 provide a good example. In addition, in Fig 6 label colours designated groups that are difficult to identify, corresponding to combinations of groups (gE- = NS1+ mock?) that are not easy to follow. Also in this section, there are reiterative paragraphs that are redundant with those in the Materials and Methods section (e.g. lines 500-505). It is unclear the real outcome of this analysis. Please re-write. Remind the above “major point 1”: with this low n º of individuals per group, the statistical significance of these results by group is null. Basically, this analysis shows that leukocyte cell counts of any type in WNV-RNA + brains is higher than in RNA – brains, reflecting that WNV loads in brains correlate with higher neuro- inflammation, mortality and severe clinical outcomes, which is expected and already shown in the other results obtained.
m. Page 20, lines 554-555: Please specify the type of IFN and TNF (Gamma? Alpha?).
n. Figure 9: It does not specify whether the results shown correspond to WT mice, TLR3KO mice or both. Please specify. Also, in Fig 9 B) lymphocyte counts are depicted, but why did the authors choose lymphocytes only? Why not monocytes too?
o. Discussion (page 22, lines 596-597): According to the text and Figure 2 A-B, mice immunized with gE alone were not tested for antibodies to NS1. Were they?
p. Id, lines 598-601. Overall, the analysis of NS1 antibodies needs some analyses to gain reliability, as suggested in “major point 2” (also applies for page 23, lines 644-5).
q. Id line 610: As remarked above (major point 2 and minor point e) there is recent evidence on the deleterious effect of NS1 immunization on protection against WNV infection in susceptible animals (birds). Therefore, the role of NS1 immunization on protection against WNV infection remains uncertain, which should be reflected in the text.
r. Id, line 611 and 633: viral loads instead viral titers.
s. Id, line 634: Figures 5E and F instead C and D.
t. Page 23, lines 649-653: Confusing sentences. Please , re-write and explain better. Take more space if needed. It is the most important finding of the work and needs more careful description.
u. Id, lines 658-660: The effect is clearly shown, but the mechanism behind it is neither explained nor speculated about. Could the authors speculate on the possible causes of this effect? Instead, may the authors provide some clues on which, in their opinion, could be the next steps to investigate the causes behind this effect?
v. Conclusions: the final sentence is very weak: the role of NS1 on WNV infection and protection is still poorly understood, so that the assertion that NS1 should be included into WNV vaccines could be wrong and risky, and in any case, more research is needed to ensure this point. Indeed, isn’t it rather odd to think on designing vaccines to protect TLR3-deficient people from WNV infection, bearing in mind that no vaccine is currently in use for ordinary people? (I ignore if this deficiency exists and, if so, which is the frequency in the population, but I guess it’s rare).
Author Response
Thank you, we appreciate your efforts.
Question | Reviewer 3 comments | Answer |
1. | Low number of individuals per group and lack of sound statistics in many of the analyses performed: Five individuals per group may be insufficient if the aim is to detect significant differences between treatments. This problem affects the whole study and can hardly be solved now. I understand the ethical constrains in numbers of animals used in experimental work, but indeed, choosing insufficient number of individuals for an experiment can be unethical if no clear conclusion over the experiment is to be obtained. I do not mean that the work is unreliable. By opposite, I believe that the main conclusions are well supported, mostly due to the thorough experimental work done using a wide range of methodologies intended to better characterize the different mechanisms displayed during WNV infection. | We agree |
But the authors need to bear this gap in mind throughout the text to avoid misinterpretations that are not formally supported statistically. For instance, survival numbers obtained in each group are somehow indicative, but there is no statistically supported difference between groups. Same for seropositive individuals, etc. Under this perspective, the word “difference” needs clarification: e.g. in the results section, paragraph 3.2.2 (“Neutralizing antibodies”), the sentence “An interesting difference…” pointing out that 3 over 5 is different from 5 over 5, the authors should think: “how many times I would expect to get this same difference if I repeat the experiment?” As experimental variation in animal models is usually high (an interesting discussion concerning experimental variation in mice inoculated with WNV can be found in Pérez-Ramírez, E. et al, JGV 2017; 98:662-670) I would suggest the authors to acknowledge this limitation in the text (especially in the discussion section) and indicate that further experimentation (with larger groups) is needed to statistically confirm the observed results. | We addressed these issues in the discussion (Lines 469 onward), including also the mentioned reference | |
2. | Anti-NS1 antibody determinations. The data obtained using the ELISA developed by the authors to detect anti-NS1 antibodies, which are presented in figure 2 A-B, are very doubtful. I do not mean the ELISA is not trustful. In fact its development is elegant and apparently correct. My concern is over the results and not over the ELISA. One issue is that the results shown in figure S4 c) barely agree with those in fig 2 A-B. In the first, almost all sera reach O.D. > 0.5 and the mean is well above this value, whereas in the second the OD values are <0.5 in almost all cases. Is there any explanation for this disagreement?
| It is true, the same amount of purified NS1 antigen induced a higher antibody response in the preliminary experiment (Fig. S4C) than in the actual immunization/challenge experiment (Fig. 2B). We cannot really explain this difference. However, this is the reason why we speak in our manuscript rather of “prior immunization with NS1” than of “antibodies against NS1”. The fact that our NS1 was actually immunogenic (able to induce antibodies) in the main experiment has been shown, though, in the combined immunizations (NS1+gE). Antigen aging or immunogenetic differences due to antigen batch properties would probably be poor explanations.
|
Indeed, the y-axes scale should be chosen to better observe these differences. I suggest choosing OD scales from 0 to 1. | Scaling in Fig. 2A,B, adapted accordingly. | |
Another issue is that the values obtained are so low that the differences between groups are very small, thus raising a question: do these small differences respond to true differences in NS1-specific antibody levels? In this regard, it is worth to remind that background noise in ELISAs may vary depending on overall serum gammaglobulin levels, which are affected by treatments such as inoculation of immune stimulators. How can the authors ensure that the small differences observed among groups are due to NS1-specific antibodies, and do not respond to different non-specific gammaglobulin levels (or other factors)? | We acknowledge in the text (paragraph 3.3.1) that the antibody response of TLR3KO mice against NS1 did not significantly differ from the reactions of the mock-immunized mice. In contrast, WT mice clearly developed antibodies against NS1. PBS/adjuvant-immunized mice served as controls for any unspecific increase of gammaglobulin levels. | |
Another recommendation for the authors at this respect is to test sera from infected mice (e.g. if obtained, those from preliminary assays to test WNV susceptibility in C57BL76 mice) on this NS1 ELISA, as well as on the DIII-gE ELISAs. These analyses will serve to two different purposes: on the one hand, to validate the ELISAs (a formal validation is lacking in this study) and, on the other hand, to compare the titres (I would strongly recommend titration in parallel) obtained in sera from infected vs immunized mice: by doing this you will have a more accurate idea on how far (or close) to the physiological Ab levels (i.e. the levels attained during a normal immune response to WNV infection) are the NS1 and gE Ab levels measured in the different experimental groups immunized with NS1, gE, NS1 plus gE, and mock. | This is a very good point. Unfortunately, we do not have enough sera left to do titrations on NS1 antigen. However, some formal validation of the tests has been done throughout the preliminary experiments (S3). Throughout these tests, the sera of gE-immunized mice had been tested not only on gE antigen but also on NS1 antigen and vice versa for NS1-immunized mice. The results were not different from the reactions of the sera from mock-vaccinated mice. In the case of gE, the ELISA results were also compared to titrations of neutralizing antibodies. | |
Note that although the group vaccinated with Equip commercial vaccine would do the job for gE-specific antibodies, this is not the case for NS1 Abs, which can hardly be stimulated by a vaccine consisting of an inactivated virus such as Equip. To finish with this point, please bear in mind that NS1 has shown to be a bad immunogen in some animal models (Rebollo, B et al, Comp Immunol Microbiol Infect Dis. 2018; 56:30-33) so there is some basis for expecting a low Ab response to NS1 in NS1-immunized individuals. | With all respect, we do not agree. The equip WNV vaccine predominantly constitutes of crude WNV-infected cell culture supernatant, which apparently contains immunogenic levels of NS1. Yet, we have taken up your suggestion, citing Rebollo et al on Line 62 . | |
3. | Effect of the time post-infection disregarded: In the analyses performed in organs the authors do not differentiate the results by the time point (dpi) they were obtained. Obviously, studies in organs arise from individuals either succumbing to the infection (i.e. during a time window of 6 to 10 dpi) or euthanized at the end of the experiment (21 dpi). This is crucial to interpret the results since, for instance, it is relatively rare to find positive viral loads in organs in survivors, while these viral loads are expected to be high in mice succumbing to the infection. This effect explains a lot of the results obtained because surviving the infection is expected to strongly correlate with low viral loads in organs, mild clinical outcomes, low inflammatory responses (cell counts, citokines, etc) in the brain and so on. For instance, in the results section 3.3.3 “Viral spread” (page 14) lines 458-460, the results are described as if the groups were homogeneous (“In the TL3KO group, WNV-RNA was detected in 3 out of 5 mice vaccinated with NS1 and all mock vaccinated mice”). The reader may legitimately wonder if those WNV-RNA positive spleens are from those mice succumbing to the infection, i.e. 3/5 from the NS1 group and 5/5 from the mock group (or otherwise). In any case, the authors should indicate this fact in the text, and probably mention this effect in the discussion. The same applies for other analyses in which the time points are important, i.e. those described under sections 3.3.3 and 3.3.4. | Indeed, this is a point to consider. We acknowledge this fact, firstly by describing the experimental conditions, lastly also by discussing the limitations of our study (Lines 569 onwards). For the interpretation of our results, we stratified according to presence or absence of WNV in the brains, which corresponded well with the clinical scores. |
4. | In the beginning of the introduction, it should be mentioned that WNV is pathogenic not only for humans and horses but also for a wide range of wild bird species. | Done, Line 38 |
5. | . “Originally from Africa” (page 1 line 38): Better “First known to Africa” since the origin of WNV remains unclear. | We have removed this sentence from the revised introduction |
6. | Id line 43: “…candidate vaccines have been extensively tested…” : please add: “in animal models”. | Done, Lines 42-43 |
7. | Page 2 line 50: non-structural proteins include NS2a, NS2b, NS4a and NS4b polypeptides. Please include. | Deleted in order to focus on NS1 and gE |
8. | Id, line 92: For NS1 there is also evidence of negative effects on protection (see Rebollo, B et al. Comp Immunol Microbiol Infect Dis. 2018; 56:30-33, above cited). | Mentioned and cited in the revised discussion |
9. | Throughout the text, the word “vaccinated” is often misused: the only true vaccinated individuals are those receiving the commercial Equip vaccine, while mice receiving doses of recombinant proteins expected to elicit an immune response should be better designated as “immunized” rather than “vaccinated”. | Corrected |
10. | Materials and Methods section: page 4 section 2.3 “Virus strains and cells”: Vero cells are mentioned twice, and the full description is not included in the first mention, but in the second. Please correct. | Corrected |
11. | Throughout the text the virus dose used in the inoculations reads “105” (one hundred and five) whereas it should read 105 (ten to the five, or one hundred thousands) (similar mistake with H2SO4 instead H2SO4 in page 5). Please correct. | Corrected |
12. | Inoculum: at several instances in the text it is stated that the inoculum injected to mice (105 pfu/mouse) is 100x LD50. However, the LD50 calculated for this strain in a previous work (cited in the text: reference #41) was 2.69 TCID50. This value is consistent with that found in another study (Pérez-Ramirez, E. et al, JGV 2017; 98:662-670, above cited) for the closely related strain IT 15803/2008 (LD50= 3.16 pfu). This means that 100 x LD50 is around 3 x102 and not 105 pfu. The authors should explain this discrepancy (perhaps linked to the mouse strain used in the experiment?). Please correct the numbers accordingly if needed. | In our experience, LD50 values may very considerably with different virus stocks. Therefore, we determined in a preliminary study the LD50 of our stocks in the context of our mice and our mode of inoculating (Results documented in supplementary data S2) |
13. | Note that in page 8 the first paragraph (“Since TLR3KO mice…”) has no section heading. It should have been numbered section 3.2, and the following sections renumbered accordingly. | Thank you. Done. |
14. | Figure 4: Panel marks (A, B, C) are lacking in the figure. Also, in the figure legend, it is unclear whether the clinical scores are cumulative, since it reads: “Each bar represents the clinical score of one individual mouse at its day of euthanasia”. Please correct to state clearly the scores are cumulative. | Figure and Legend corrected accordingly |
15. | Section 3.3.4 (Page 16): Flow cytometry results are presented in a most confusing way: Lines 509-512 provide a good example. In addition, in Fig 6 label colours designated groups that are difficult to identify, corresponding to combinations of groups (gE- = NS1+ mock?) that are not easy to follow. Also in this section, there are reiterative paragraphs that are redundant with those in the Materials and Methods section (e.g. lines 500-505). It is unclear the real outcome of this analysis. Please re-write. Remind the above “major point 1”: with this low n º of individuals per group, the statistical significance of these results by group is null. Basically, this analysis shows that leukocyte cell counts of any type in WNV-RNA + brains is higher than in RNA – brains, reflecting that WNV loads in brains correlate with higher neuro- inflammation, mortality and severe clinical outcomes, which is expected and already shown in the other results obtained. | The complete set of data, from each single mouse has been added in the form of a supplementary Excel Table S1 (Supplementary data 4). Reiterations removed. |
16. | Page 20, lines 554-555: Please specify the type of IFN and TNF (Gamma? Alpha?). | Done |
17. | Figure 9: It does not specify whether the results shown correspond to WT mice, TLR3KO mice or both. Please specify. | Specified in the Legend to Figure 9 |
Also, in Fig 9 B) lymphocyte counts are depicted, but why did the authors choose lymphocytes only? Why not monocytes too? | The various inflammatory cell populations were similarly independent of the viral titers. We show lymphocytes as a representative example (specified in the revised text) | |
18. | Discussion (page 22, lines 596-597): According to the text and Figure 2 A-B, mice immunized with gE alone were not tested for antibodies to NS1. Were they? | Indeed, the non-vaccinating antigen was consistently used as a control for the vaccinating antigen. None of the animals immunized with a single antigen developed reactivity (antibodies) against the non-vaccinating antigen. Mentioned in the revised Materials and Methods. |
19. | Id, lines 598-601. Overall, the analysis of NS1 antibodies needs some analyses to gain reliability, as suggested in “major point 2” (also applies for page 23, lines 644-5). | Cross-reaction and unspecific reaction were at a negligible low level. |
20. | Id line 610: As remarked above (major point 2 and minor point e) there is recent evidence on the deleterious effect of NS1 immunization on protection against WNV infection in susceptible animals (birds). Therefore, the role of NS1 immunization on protection against WNV infection remains uncertain, which should be reflected in the text. | We did not recognize any adverse effects upon immunizing with NS1 (inserted and referenced in the revised Discussion). |
21. | Id, line 611 and 633: viral loads instead viral titers. | Corrected systematically |
22. | Id, line 634: Figures 5E and F instead C and D. | Thank you. Corrected. |
23. | Page 23, lines 649-653: Confusing sentences. Please , re-write and explain better. Take more space if needed. It is the most important finding of the work and needs more careful description. | Thank you. We revised the entire Discussion, taking this suggestion very seriously. Paragraph re-written accordingly. |
24. | . Id, lines 658-660: The effect is clearly shown, but the mechanism behind it is neither explained nor speculated about. Could the authors speculate on the possible causes of this effect? Instead, may the authors provide some clues on which, in their opinion, could be the next steps to investigate the causes behind this effect? | Included in the revised Discussion. |
25. | Conclusions: the final sentence is very weak: the role of NS1 on WNV infection and protection is still poorly understood, so that the assertion that NS1 should be included into WNV vaccines could be wrong and risky, and in any case, more research is needed to ensure this point. Indeed, isn’t it rather odd to think on designing vaccines to protect TLR3-deficient people from WNV infection, bearing in mind that no vaccine is currently in use for ordinary people? (I ignore if this deficiency exists and, if so, which is the frequency in the population, but I guess it’s rare). | We intended to EXTEND possible protection against WNV to people with defects in the innate immune system, rather than to invent a vaccine specifically for those. However, we strongly advocate testing of such vaccines in non-human primates prior to use in humans. We included both of these thoughts into the revised discussion. |

Round 2
Reviewer 2 Report
I appreciate the authors’ thorough revisions and response to comments. This iteration of the manuscript is easier to understand. The introduction is markedly improved. I have very minor suggestions/questions before accepting this manuscript:
In the discussion:
Remove “of” from line 897 (despite of some..).
This is much easier to read, but some additional questions. You discuss that animal numbers may play a role in your ability to detect differences in brain invasion between groups, but you are comparing your results to TLR3 studies primarily done using NY99 strains. How do you think this affects your results? How different are these strains?
In the conclusions, the statement “WNV brain invasion was the major trigger of encephalitis in unprotected…mice” is still problematic. It is well established that viral invasion leads to encephalitis and this does not highlight the important and novel conclusions of this study. I would suggest the authors start the conclusions with the findings stated in the second sentence and reword. Perhaps try: "The degree of inflammation measured in the brain was affected by the viral load, TLR3 signaling, and previous immunization again NS1...."
Author Response
Thank you, we have followed your advise as stated in the attached table.
1. I appreciate the authors’ thorough
revisions and response to comments. This
iteration of the manuscript is easier to
understand. The introduction is markedly
improved.
Thank you
2. I have very minor suggestions/questions
before accepting this manuscript:
In the discussion:
Remove “of” from line 897 (despite of
some..).
Done
3. This is much easier to read, but some
additional questions. You discuss that
animal numbers may play a role in your
ability to detect differences in brain
invasion between groups, but you
are comparing your results to TLR3
studies primarily done using NY99
strains. How do you think this affects your
results? How different are these strains?
Ita09 and NY99 share
more than 99% identity
on the amino acid level;
somewhat less on the
nucleotide level. We
cannot know how these
would influence the
results of our study.
To acknowledge this, we
have rephrased the last
sentence of our
Discussion to mention
the need of testing our
findings with various
WNV strains, i.e.:
“While further studies
using various WNV
strains in appropriate
animal models,
particularly non-human
primates, are warranted
…”
4. In the conclusions, the statement “WNV
brain invasion was the major trigger of
encephalitis in unprotected…mice” is still
problematic. It is well established that
viral invasion leads to encephalitis and
this does not highlight the important and
novel conclusions of this study. I would
suggest the authors start the conclusions
with the findings stated in the second
sentence and reword. Perhaps try: "The
degree of inflammation measured in the
brain was affected by the viral load, TLR3
signaling, and previous immunization
again NS1...."
Done as suggested
5. Spell check required
11 spelling errors corrected

Reviewer 3 Report
The authors have answered satisfactorily to all my queries.
Author Response
Thank you. We appreciate your efforts.
